# TokenFlow: Consistent Diffusion Features for Consistent Video Editing

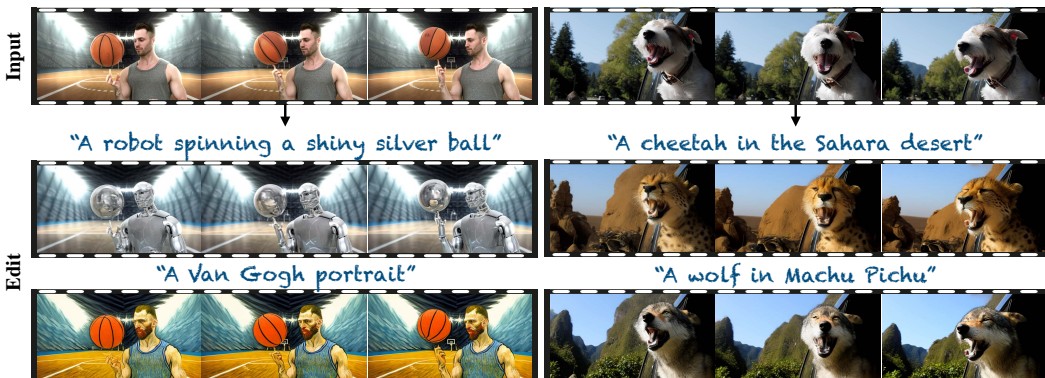

Figure 1: TokenFlow enables consistent, high-quality semantic edits of real-world videos. Given an input video (top row), our method edits it according to a target text prompt (middle and bottom rows), while preserving the semantic layout and motion in the original scene.

## Abstract

The generative AI revolution has recently expanded to videos. Nevertheless, current state-of-the-art video models are still lagging behind image models in terms of visual quality and user control over the generated content. In this work, we present a framework that harnesses the power of a text-to-image diffusion model for the task of text-driven video editing. Specifically, given a source video and a target text-prompt, our method generates a high-quality video that adheres to the target text, while preserving the spatial layout and motion of the input video. Our method is based on a key observation that consistency in the edited video can be obtained by enforcing consistency in the diffusion feature space. We achieve this by explicitly propagating diffusion features based on inter-frame correspondences, readily available in the model. Thus, our framework does not require any training or fine-tuning, and can work in conjunction with any off-the-shelf text-to-image editing method. We demonstrate state-of-the-art editing results on a variety of real-world videos.

## 1 Introduction

The evolution of text-to-image models has recently facilitated advances in image editing and content creation, allowing users to control various proprieties of both generated and real images. Nevertheless, expanding this exciting progress to video is still lagging behind. A surge of large-scale text-to-video generative models has emerged, demonstrating impressive results in generating clips solely from textual descriptions. However, despite the progress made in this area, existing video models are still in their infancy, being limited in resolution, video length, or the complexity of video dynamics they can represent. In this paper, we harness the power of a state-of-the-art pre-trained text-to-*image* model for the task of text-driven editing of natural videos. Specifically, our goal is to generate high-quality videos that adhere to the target edit expressed by an input text prompt, while preserving the spatial layout and motion of the original video. The main challenge in leveraging an image diffusion model for video editing is to ensure that the edited content is consistent across all video frames – ideally, each physical point in the 3D world undergoes coherent modifications across time. Existing and concurrent video editing methods that are based on image diffusion models have demonstrated that global appearance coherency across the edited frames can be achieved by extending the self-attention module to include multiple frames (Wu et al., 2022; Khachatryan et al., 2023b; Ceylan et al., 2023; Qi et al., 2023). Nevertheless, this approach is insufficient for achieving the desired level of temporal consistency, as motion in the video is only *implicitly* preserved through the

attention module. Consequently, professionals or semi-professionals users often resort to elaborate video editing pipelines that entail additional manual work. In this work, we propose a framework to tackle this challenge by *explicitly* enforcing the original inter-frame correspondences on the edit. Intuitively, natural videos contain redundant information across frames, e.g., depict similar appearance and shared visual elements. Our key observation is that the internal representation of the video in the diffusion model exhibits similar properties. That is, the level of redundancy and temporal consistency of the frames in the RGB space and in the diffusion feature space are tightly correlated. Based on this observation, the pillar of our approach is to achieve consistent edit by ensuring that the features of the edited video are consistent across frames. Specifically, we enforce that the edited features convey the same inter-frame correspondences and redundancy as the original video features. To do so, we leverage the original inter-frame feature correspondences, which are readily available by the model. This leads to an effective method that directly propagates the *edited* diffusion features based on the *original* video dynamics. This approach allows us to harness the generative prior of state-of-the-art image diffusion model without additional training or fine-tuning, and can work in conjunction with an off-the-shelf diffusion-based image editing method (e.g., Meng et al. (2022); Hertz et al. (2022); Zhang & Agrawala (2023); Tumanyan et al. (2023)).

To summarize, we make the following key contributions:

- A technique, dubbed *TokenFlow*, that enforces semantic correspondences of diffusion features across frames, allowing to significantly increase temporal consistency in videos generated by a text-to-image diffusion model.

- Novel empirical analysis studying the proprieties of diffusion features across a video.

- State-of-the-art editing results on diverse videos, depicting complex motions.

## 2 RELATED WORK

**Text-driven image & video synthesis**   Seminal works designed GAN architectures to synthesize images conditioned on text embeddings (Reed et al., 2016; Zhang et al., 2016). With the ever-growing scale of vision-language datasets and pretraining strategies (Radford et al., 2021; Schuhmann et al., 2022), there has been a remarkable progress in text-driven image generation capabilities. Users can sytnesize high-quality visual content using simple text prompts. Much of this progress is also attributed to diffusion models (Sohl-Dickstein et al., 2015; Croitoru et al., 2022; Dhariwal & Nichol, 2021; Ho et al., 2020; Nichol & Dhariwal, 2021) which have been established as state-of-the-art text-to-image generators (Nichol et al., 2021; Saharia et al., 2022; Ramesh et al., 2022; Rombach et al., 2022; Sheynin et al., 2022; Bar-Tal et al., 2023). Such models have been extended for text-to-video generation, by extending 2D architectures to the temporal dimension (e.g., using temporal attention Ho et al. (2022b)) and performing large-scale training on video datasets (Ho et al., 2022a; Blattmann et al., 2023; Singer et al., 2022). Recently, Gen-1 (Esser et al., 2023) tailored a diffusion model architecture for the task of video editing, by conditioning the network on structure/appearance representations. Nevertheless, due to their extensive computation and memory requirements, existing video diffusion models are still in infancy and are largely restricted to short clips, or exhibit lower visual quality compared to image models. On the other side of the spectrum, a promising recent trend of works leverage a pre-trained image diffusion model for video synthesis tasks, without additional training (Fridman et al., 2023; Wu et al., 2022; Lee et al., 2023a; Qi et al., 2023). Our work falls into this category, employing a pretrained text-to-image diffusion model for the task of video editing, without any training or finetuning.

**Consistent video stylization**   A common approach for video stylization involves applying image editing techniques (e.g., style transfer) on a frame-by-frame basis, followed by a post-processing stage to address temporal inconsistencies in the edited video (Lai et al. (2018b); Lei et al. (2020; 2023)). Although these methods effectively reduce high-frequency temporal flickering, they are not designed to handle frames that exhibit substantial variations in content, which often occur when applying text-based image editing techniques (Qi et al., 2023). Kasten et al. (2021) propose to decompose a video into a set of 2D atlases, each provides a unified representation of the background or of a foreground object throughout the video. Edits applied to the 2D atlases are automatically mapped back to the video, thus achieving temporal consistency with minimal effort. Bar-Tal et al. (2022); Lee et al. (2023b) leverage this representation to perform text-driven editing. However, the atlas representation is limited to videos with simple motion and requires long training, limiting the applicability of this technique and of the methods built upon it. Our work is also related to classical works that demonstrated that small patches in a natural video extensively repeat across frames (Shahar et al., 2011; Cheung et al., 2005), and thus consistent editing can by simplified by editing a subset of keyframes and propagating the edit across the video by establishing patch correspondences using handcrafted features and optical flow (Ruder et al., 2016; Jamriška et al., 2019) or by

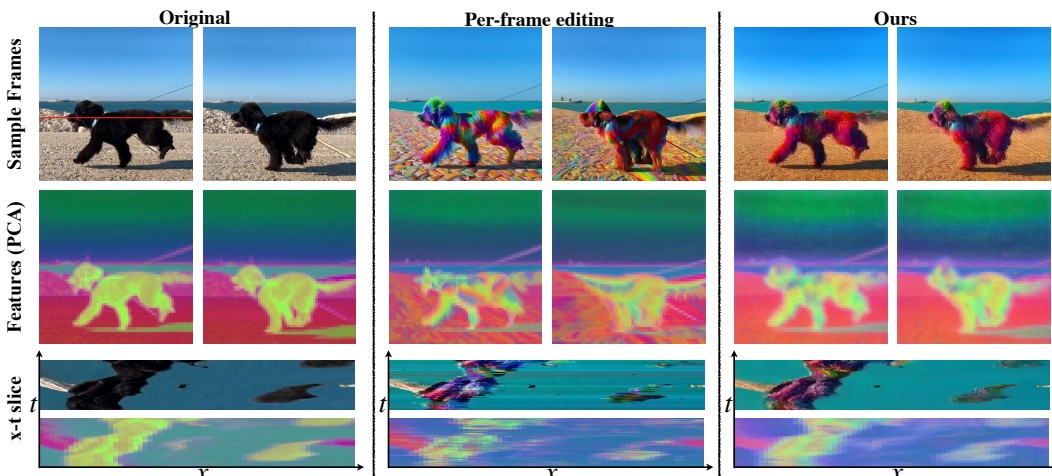

Figure 3: **Diffusion features across time.** *Left:* Given an input video (top row), we apply DDIM inversion on each frame and extract features from the highest resolution decoder layer in $\epsilon_\theta$. We apply PCA on the features (i.e., output tokens from the self-attention module) extracted from all frames and visualize the first three components (second row). We further visualize an $x$-$t$ slice (marked in red on the original frame) for both RGB and features (bottom row). The feature representation is consistent across time – corresponding regions are encoded with similar features across the video. *Middle:* Frames and feature visualization for an edited video obtained by applying an image editing method (Tumanyan et al. (2023)) on each frame; inconsistent patterns in RGB are also evident in the feature space (e.g., on the dog's body). *Right:* Our method enforces the edited video to convey the same level of feature consistency as the original video, which translates into a coherent and high-quality edit in RGB space.

training a patch-based GAN (Texler et al., 2020). Nevertheless, such propagation methods struggle to handle videos with illumination changes, or with complex dynamics. Importantly, they rely on a user provided consistent edit of the keyframes, which remains a labor-intensive task yet to be automated. Yang et al. (2023) combines keyframe editing with a propagation method by Jamriška et al. (2019). They edit keyframes using a text-to-image diffusion model while enforcing optical flow constraints on the edited keyframes. However, since optical flow estimation between distant frames is not reliable, their method fails to consistently edit keyframes that are far apart (as seen in our Supplementary Material - SM), and as a result, fails to consistently edit most videos.

Our work shares a similar motivation as this approach that benefits from the temporal redundancies in natural videos. We show that such redundancies are also present in the feature space of a text-to-image diffusion model, and leverage this property to achieve consistency.

**Controlled generation via diffusion features manipulation** Recently, a surge of works demonstrated how text-to-image diffusion models can be readily adapted to various editing and generation tasks, by performing simple operations on the intermediate feature representation of the diffusion network (Chefer et al., 2023; Hong et al., 2022; Ma et al., 2023; Tumanyan et al., 2023; Hertz et al., 2022; Patashnik et al., 2023; Cao et al., 2023). Luo et al. (2023); Zhang et al. (2023) demonstrated semantic appearance swapping using diffusion feature correspondences. Hertz et al. (2022) observed that by manipulating the cross-attention layers, it is possible to control the relation between the spatial layout of the image to each word in the text. Plug-and-Play Diffusion (PnP, Tumanyan et al. (2023)) analyzed the spatial features and the self-attention maps and found that they capture semantic information at high spatial granularity. Tune-A-Video (Wu et al., 2022) observed that by extending the self-attention module to operate on more than a sin-

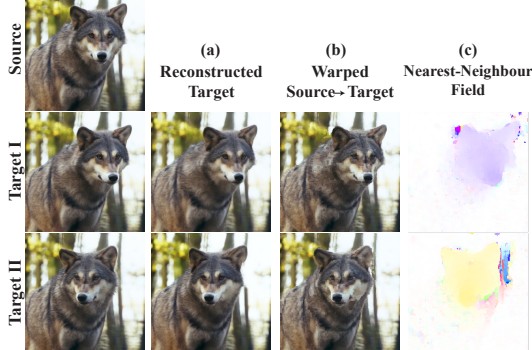

Figure 2: **Fine-grained feature correspondences.** Features (i.e., output tokens from the self-attention modules) extracted from of a source frame are used to reconstruct nearby frames. This is done by: (a) swapping each feature in the target by its nearest feature in the source, in all layers and all generation time steps, and (b) simple warping in RGB space, using a nearest neighbour field (c), computed between the source and target features extracted from the highest resolution decoder layer. The target is faithfully reconstructed, demonstrating the high level of spatial granularity and shared content between the features.

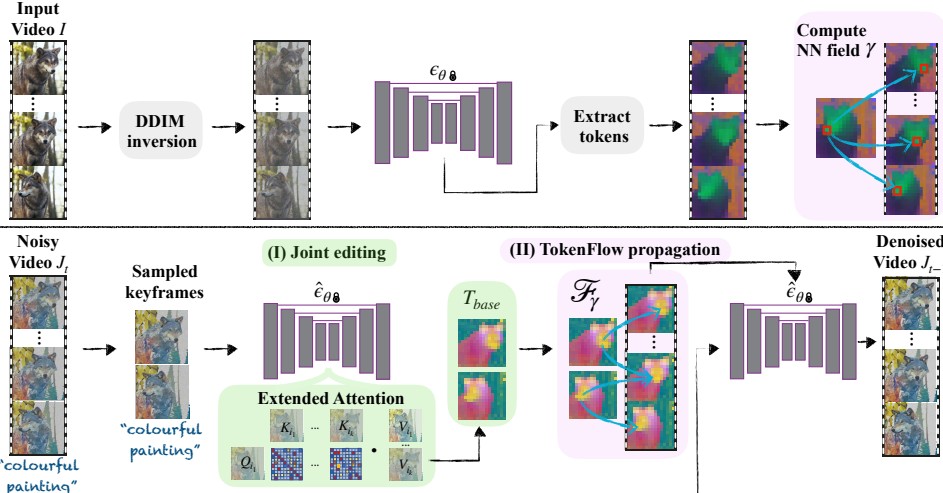

Figure 4: **TokenFlow pipeline**. Top: Given an input video $\mathcal{I}$, we DDIM invert each frame, extract its tokens, i.e., output features from the self-attention modules, from each timestep and layer, and compute inter-frame features correspondences using a nearest-neighbor (NN) search. Bottom: The edited video is generated as follows: at each denoising step $t$, (I) we sample keyframes from the noisy video $J_t$ and jointly edit them using an extended-attention block; the set of resulting edited tokens is $\mathbf{T}_{base}$. (II) We propagate the edited tokens across the video according to the pre-computed correspondences of the original video features. To denoise $J_t$, we feed each frame to the network, and replace the generated tokens with the tokens obtained from the propagation step (II).

gle frame, it is possible to generate frames that share a common global appearance. Qi et al. (2023); Ceylan et al. (2023); Khachatryan et al. (2023a); Shin et al. (2023); Liu et al. (2023) leverage this property to achieve globally-coherent video edits. Nevertheless, as demonstrated in Sec. 5, inflating the self-attention module is insufficient for achieving fine-grained temporal consistency. Prior and concurrent works either compromise visual quality, or exhibit limited temporal consistency. In this work, we also perform video editing via simple operations in the feature space of a pre-trained text-to-image model, we explicitly encourage the features of the model to be temporally consistent through *TokenFlow*.

## 3 PRELIMINARIES

**Diffusion Models**   Diffusion probabalistic models (DPM) (Sohl-Dickstein et al., 2015; Croitoru et al., 2022; Dhariwal & Nichol, 2021; Ho et al., 2020; Nichol & Dhariwal, 2021) are a class of generative models that aim to approximate a data distribution $q$ through a progressive denosing process. Starting from a Gaussian i.i.d noisy image $x_T \sim \mathcal{N}(0, I)$, the diffusion model $\epsilon_\theta$, gradually denoises it, until reaching a clean image $x_0$ drawn from the target distribution $q$. DPM can learn a conditional distribution by incorporating additional guiding signals, such as text conditioning.
Song et al. (2020) derived DDIM, a deterministic sampling algorithm given an initial noise $x_T$. By applying this algorithm in the reverse order (a.k.a. DDIM inversion) starting from the clean $x_0$, it allows to obtain the intermediate noisy images $\{x_i\}_{t=1}^{T}$ used to generate it.

**Stable Diffusion**   Stable Diffusion (SD) (Rombach et al., 2022) is a prominent text-to-image diffusion model that operates in a latent image space. A pretrained encoder maps RGB images to this space, and a decoder decodes latents back to high-resolution images. In more detail, SD is based on a U-Net architecture (Ronneberger et al., 2015), which comprises of residual, self-attention, and cross-attention blocks. The residual block convolves the activations from a previous layer, while cross-attention manipulates features according to the text prompt. In the self-attention block, features are projected into queries $Q$, keys $K$, and values $V$. The `Attention` operation (Vaswani et al., 2017) computes the affinities between the $d$-dimensional projections $Q, K$ to yield the output of the layer:

$$A \cdot V \text{ where } A = \texttt{Attention}(Q; K) \text{ and } \texttt{Attention}(Q; K) = \texttt{Softmax}\left(\frac{QK^T}{\sqrt{d}}\right) \quad (1)$$

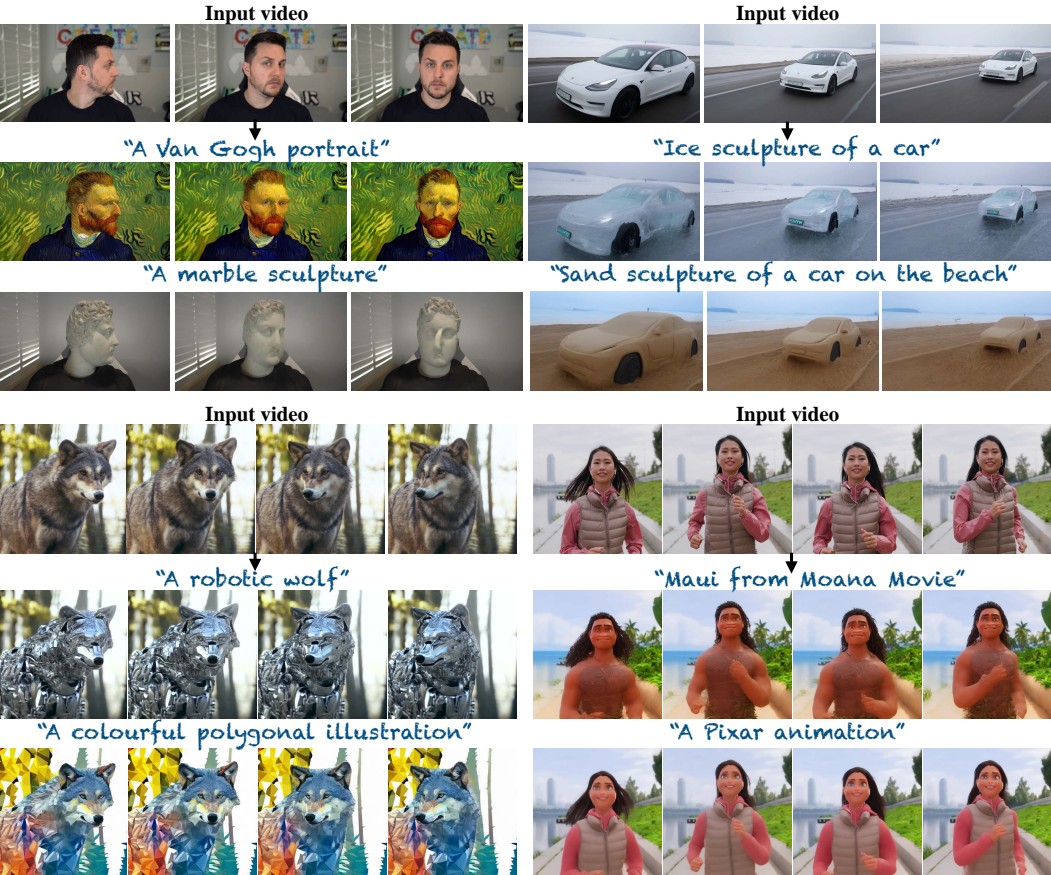

Figure 5: **Results.** Sample results of our method. We refer the reader to our webpage and SM for more examples and full-video results.

## 4 METHOD

Given an input video $\mathcal{I} = [\boldsymbol{I}^1, ..., \boldsymbol{I}^n]$, and a text prompt $\mathcal{P}$ describing the target edit, our goal is to generate an edited video $\mathcal{J} = [\boldsymbol{J}^1, ..., \boldsymbol{J}^n]$ that adheres to the text $\mathcal{P}$, while preserving the original motion and semantic layout of $\mathcal{I}$. To achieve this, our framework leverages a pretrained and fixed text-to-*image* diffusion model $\epsilon_\theta$.

Naïvely leveraging $\epsilon_\theta$ for *video* editing, by applying an image editing method on each frame independently (e.g., Hertz et al. (2022); Tumanyan et al. (2023); Meng et al. (2022); Zhang & Agrawala (2023)), results in content inconsistencies across frames (e.g., Fig. 3 middle column). Our key finding is that these inconsistencies can be alleviated by enforcing consistency among the internal diffusion features across frames, during the editing process.

Natural videos typically depict coherent and shared content across time. We observe that the internal representation of natural videos in $\epsilon_\theta$ has similar properties. This is illustrated in Fig. 3, where we visualize the features extracted from a given video (first column). As seen, the features depict a shared and consistent representation across frames, i.e., corresponding regions exhibit similar representation. We further observe that the original video features provide fine-grained correspondences between frames, using a simple nearest neighbour search (Fig 2). Moreover, we show that these *corresponding features are interchangeable for the diffusion model* – we can faithfully synthesize one frame by swapping its features by their corresponding ones in a nearby frame (Fig 2(a)).

Nevertheless, when an edit is applied to each frame individually, the consistency of the features breaks (Fig. 3 middle column). This implies that the level of consistency of in RGB space is correlated with the consistency of the internal features of the frames. Hence, our key idea is to manipulate the features of the edited video to preserve the level of consistency and inter-frame correspondences of the original video features.

As illustrated in Fig. 4, our framework, dubbed *TokenFlow*, alternates at each generation timestep between two main components: (i) sampling a set of keyframes and jointly editing them according to $\mathcal{P}$; this stage results in shared global appearance across the keyframes, and (ii) propagating the features from the keyframes to all of the frames based on the correspondences provided by the original

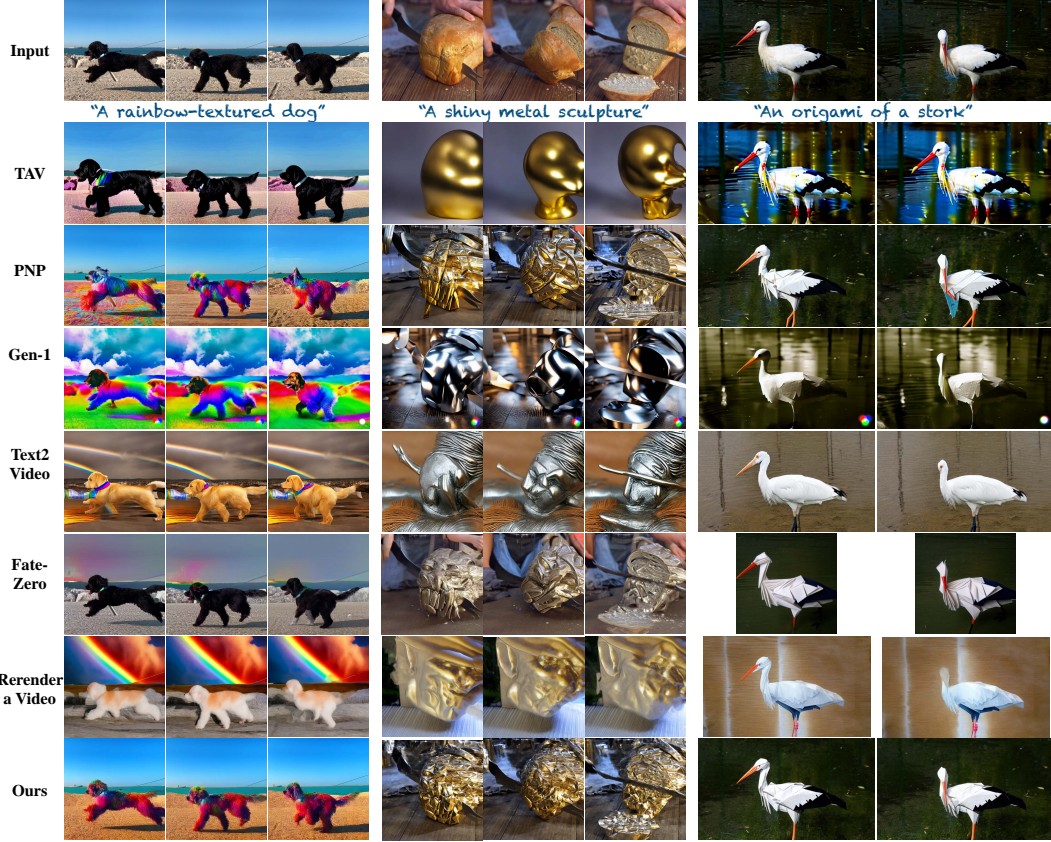

Figure 6: **Comparison.** We compare our method against Tune-A-Video (TAV, Wu et al. (2022)), PnP-Diffusion (Tumanyan et al., 2023) applied per frame, Gen-1 (Esser et al., 2023), Text2Video-Zero (Khachatryan et al., 2023a) and Fate-Zero (Qi et al., 2023). We refer the reader to our supplementary material for full-video comparisons.

video features; this stage explicitly preserves the consistency and fine-grained shared representation of the original video features. Both stages are done in combination with an image editing technique $\hat{\epsilon}_\theta$ (e.g, Tumanyan et al. (2023)). Intuitively, the benefit of alternating between keyframe editing and propagation is twofold: first, sampling random keyframes at each generation step increases the robustness to a particular selection. Second, since each generation step results in more consistent features, the sampled keyframes in the next step will be edited more consistently.

**Pre-processing: extracting diffusion features.** Given an input video $\mathcal{I}$, we apply DDIM inversion (see Sec. 3) on each frame $\boldsymbol{I}^i$, which yields a sequence of latents $[\boldsymbol{x}_1^i, ..., \boldsymbol{x}_T^i]$. For each generation timestep $t$, we feed the latent $\boldsymbol{x}_t^i$ of each frame $i \in [n]$ to the model and extract the tokens $\phi(x_t^i)$ from the self-attention module of every layer in the network $\epsilon_\theta$ (fig. 4, top). We will later use these tokens to establish inter-frame correspondences between diffusion features.

### 4.1 KEYFRAME SAMPLING AND JOINT EDITING

Our observations imply that given the features of a single edited frame, we can generate the next frames by propagating its features to their corresponding locations. Most videos, however, can not be represented by a single keyframe. To account for that, we consider multiple keyframes, from which we obtain a set of features (tokens), $\boldsymbol{T}_{base}$, that will later be propagated to the entire video. Specifically, at each generation step, we randomly sample a set of keyframes $\{\boldsymbol{J}^i\}_{i \in \kappa}$ in fixed frame intervals (see SM for details). We joinly edit the keyframes by extending the self-attention block to simultaneously process them (Wu et al., 2022), thus encouraging them to share a global appearance. In more detail, the input to the modified block are the self-attention features from all keyframes $\{\boldsymbol{Q}^i\}_{i \in \kappa}, \{\boldsymbol{K}^i\}_{i \in \kappa}, \{\boldsymbol{V}^i\}_{i \in \kappa}$ where $\boldsymbol{Q}^i, \boldsymbol{K}^i, \boldsymbol{V}^i$ are the queries, keys, and values of frame $i \in \kappa, \kappa = \{i_1, ...i_k\}$. The keys of all frames are concatenated, and the extended-attention is:

$$\texttt{ExtAttn}\Big(\boldsymbol{Q}^i; [\boldsymbol{K}^{i_1}, ... \boldsymbol{K}^{i_k}]\Big) = \texttt{Softmax}\left(\frac{\boldsymbol{Q}^i \left[\boldsymbol{K}^{i_1}, ... \boldsymbol{K}^{i_k}\right]^T}{\sqrt{d}}\right) \tag{2}$$

The output of the block for frame $i$ is given by:

$$\phi(\boldsymbol{J}^i) = \hat{\boldsymbol{A}} \cdot [\boldsymbol{V}^{i_1}, \dots \boldsymbol{V}^{i_k}] \quad \text{where} \quad \hat{\boldsymbol{A}} = \texttt{ExtAttn}\Big(\boldsymbol{Q}^i; [\boldsymbol{K}^{i_1}, \dots \boldsymbol{K}^{i_k}]\Big) \tag{3}$$

Intuitively, each keyframe queries all other keyframes, and aggregates information from them. This results in a roughly unified appearance in the edited frames (Wu et al., 2022; Khachatryan et al., 2023b; Ceylan et al., 2023; Qi et al., 2023). We define $\mathbf{T}_{base} = \{\phi(\boldsymbol{J}^i)\}_{i \in \kappa}$, for each layer in the network (Fig. 4 bottom middle).

### 4.2 Edit Propagation via TokenFlow

Given $\mathbf{T}_{base}$, we propagate it across the video based on the token correspondences extracted from the original video. At each generation step $t$, we compute the nearest neighbor (NN) of each original frame's tokens, $\phi(x_t^i)$, and its two adjacent keyframes' tokens, $\phi(x_t^{i+})$, $\phi(x_t^{i-})$ where $i+$ is the index of the closest future keyframe, and $i-$ the index of the closest past keyframe. Denote the resulting NN fields $\gamma^{i+}, \gamma^{i-}$:

$$\gamma^{i\pm}[p] = \arg\min_q \mathcal{D}\left(\phi(\boldsymbol{x}^i)[p], \phi(\boldsymbol{x}^{i\pm})[q]\right) \tag{4}$$

Where $p, q$ are spatial locations in the token feature map, and $\mathcal{D}$ is cosine distance. For simplicity, we omit the generation timestep $t$; our method is applied in all time-steps and self-attention layers. Once we obtain $\gamma^{\pm}$, we use it to propagate the *edited* frames' tokens $\mathbf{T}_{base}$ to the rest of the video, by linearly combining the tokens in $\mathbf{T}_{base}$ corresponding to each spatial location $p$ and frame $i$:

$$\mathcal{F}_\gamma(\mathbf{T}_{base}, i, p) = w_i \cdot \phi(\boldsymbol{J}^{i+})[\gamma^{i+}[p]] + (1 - w_i) \cdot \phi(\boldsymbol{J}^{i-})[\gamma^{i-}[p]] \tag{5}$$

Where $\phi(\boldsymbol{J}^{i\pm}) \in \mathbf{T}_{base}$ and $w_i \in (0, 1)$ is a scalar proportional to the distance between frame $i$ and its adjacent keyframes (see SM), ensuring a smooth transition. Note that $\mathcal{F}$ also modifies the tokens of the sampled keyframes. That is, we modify the self-attention blocks to output a linear combination of the tokens in $\mathbf{T}_{base}$ for all frames, including the keyframes, according to the original video token correspondences.

**Overall algorithm** We summarize our video editing algorithm in Alg. 1: We first perform DDIM inversion on the input video $\mathcal{I}$ and extract the sequence of noisy latents $\{x_t^i\}_{t=1}^T$ for all frames $i \in [n]$ (fig 4, top). We then denoise the video, alternating between keyframes editing and *TokenFlow* propagation: At each generation step $t$, we randomize $k < n$ keyframe indices, and denoise them using an image editing technique (e.g., Tumanyan et al. (2023); Meng et al. (2022); Zhang & Agrawala (2023)) combined with extended-attention (Eq. 3, Fig. 4 (I)). We then denoise the entire video $\mathcal{J}_t$ by combining the image-editing technique with *TokenFlow* (Eq. 5, Fig. 4 (II))

---

**Algorithm 1** TokenFlow editing

**Input:**

$\quad \mathcal{I} = [\mathbf{I}^1, ..., \mathbf{I}^n]$ ▷ Input Video
$\quad \boldsymbol{P}$ ▷ Target text prompt
$\quad \hat{\boldsymbol{\Psi}}$ ▷ Diffusion-based image editing technique
$\{\mathbf{x}_t^i\}_{t=1}^T, \{\phi(x_i)\}_{i=1}^n \leftarrow \text{DDIM-Inv}[\mathbf{I}^i] \quad \forall i \in [n], \; t \in [T]$
$\mathbf{J}_T^1, \dots, \mathbf{J}_T^n \leftarrow \mathbf{x}_T^1, \dots, \mathbf{x}_T^n$
**For** $t = T, \dots, 1$ **do**
$\quad \mathcal{K} = \{i_1, \dots, i_k\} \leftarrow$ sample keyframe indices
$\quad \mathcal{F}_\gamma \leftarrow \gamma^{i\pm} \; \forall i \in [n] \quad$ compute NN field
$\quad \{\mathbf{J}_{t-1}^j\}_{j \in \mathcal{K}} \leftarrow \hat{\epsilon}_\theta[\{\mathbf{J}_t^j\}_{j \in \mathcal{K}}; \texttt{ExtAttn}]$
$\quad \mathbf{T}_{\text{base}} \leftarrow \phi(\{\mathbf{J}_{t-1}^j\}_{j \in \mathcal{K}}) \quad$ extract keyframes' tokens
$\quad \mathbf{J}_{t-1} \leftarrow \hat{\epsilon}_\theta[\mathbf{J}_t; \text{TokenFlow}(\mathcal{F}_\gamma(\mathbf{T}_{\text{base}}))]$
**Output:** $\mathcal{J} = [\mathbf{J}_0^1, \dots, \mathbf{J}_0^n]$

---

at every self-attention block in every layer of the network. Note that each layer includes a residual connection between the input and output of the self-attention block, thus performing *TokenFlow* at each layer is necessary.

## 5 Results

We evaluate our method on DAVIS videos (Pont-Tuset et al., 2017) and on Internet videos depicting animals, food, humans, and various objects in motion. The spatial resolution of the videos is $384 \times 672$ or $512 \times 512$ pixels, and they consist of 40 to 200 frames. We use various text prompts on each video to obtain diverse editing results. Our evaluation dataset comprises of 61 text-video pairs. We utilize PnP-Diffusion (Tumanyan et al., 2023) as the frame editing method, and we use the same hyper-parameters for all our results. PnP-Diffusion may fail to accurately preserve the structure of each frame due to inaccurate DDIM inversion (see Fig. 3, middle column, right frame: the dog's head is distorted). Our method improves robustness to this, as multiple frames contribute to the generation of each frame in the video. Our framework can be combined with any diffusion-based image editing technique that accurately preserves the structure of the images; results with different

image editing techniques (e.g. Meng et al. (2022); Zhang & Agrawala (2023)) are available in the SM. Fig. 5 and 1 show sample frames from the edited videos. Our edits are temporally consistent and adhere to the edit prompt. The man's head is changed to Van-Gogh or marble (top left); importantly, the man's identity and the scene's background are consistent throughout the video. The patterns of the polygonal wolf (bottom left) are the same across time: the body is *consistently* orange while the chest is blue. We refer the reader to the SM for implementation details and video results.

**Baselines.** We compare our method to state-of-the-art, and concurrent works: (i) Fate-Zero (Qi et al., 2023) and (ii) Text2Video-Zero (Khachatryan et al., 2023b), that utilize a text-to-image model for video editing using self-attention inflation. (iii) Re-render a Video (Yang et al., 2023) that edits keyframes by adding optical flow optimization to self-attention inflation of an image model, and then propagates the edit from the keyframes to the rest of the video using an off-the-shelf propagation method. (iv) Tune-a-Video (Wu et al., 2022) that fine-tunes the text-to-image model on the given test video. (v) Gen-1 (Esser et al., 2023), a video diffusion model that was trained on a large-scale image and video dataset. (vi) Per-frame diffusion-based image editing baseline, PnP-Diffusion (Tumanyan et al., 2023). We additionally consider the two following baselines: (i) Text2LIVE (Bar-Tal et al., 2022) which utilize a layered video representation (NLA) (Kasten et al., 2021) and perform test-time training using CLIP losses. Note that NLA requires foreground/background separation masks and takes ∼ 10 hours to train. (ii) Applying PnP-Diffusion on a single keyframe and propagating the edit to the entire video using Jamriška et al. (2019).

## 5.1 QUALITATIVE EVALUATION

Fig. 6 provides a qualitative comparison of our method to prominent baselines; please refer to SM for the full videos. Our method (bottom row) outputs videos that better adhere to the edit prompt while maintaining temporal consistency of the resulting edited video, while other methods struggle to meet both these goals. Tune-A-Video (second row) inflates the 2D image model into a video model, and fine-tunes it to overfit the motion of the video; thus, it is suitable for short clips. For long videos it struggles to capture the motion resulting with meaningless edits, e.g., the shiny metal sculpture. Applying PnP for each frame independently (third row) results in exquisite edits adhering to the edit prompt but, as expected, lack any temporal consistency. The results of Gen-1 (fourth row) also suffer from some temporal inconsistencies (the beak of the origami stork changes color). Moreover, their frame quality is significantly worse than that of a text-to-image diffusion model. The edits of Text2Video-Zero and Fate-Zero (fifth and sixth row) suffer from severe jittering as these methods rely heavily on the extended attention mechanism to *implicitly* encourage consistency. The results of Rerender-a-Video exhibit notable long-range inconsistencies and artifacts arising primarily from their reliance on optical flow estimation for distant frames (e.g. keyframes), which is known to be sub-optimal (See our video results in the SM; when the wolf turns its head, the nose color changes). We provide qualitative comparison to Text2LIVE and to a RGB propagation baseline in the SM.

## 5.2 QUANTITATIVE EVALUATION

We evaluate our method in terms of: (i) *edit fidelity* measured by computing the average similarity between the CLIP embedding (Radford et al., 2021) of each edited frame and the target text prompt; (ii) *temporal consistency*. Following Ceylan et al. (2023); Lai et al. (2018a), temporal consistency is measured by (a) computing the optical flow of the original video using Teed & Deng (2020), warping the edited frames according to it, and measuring the warping error, and (b) a user study; We adopt a Two-alternative Forced Choice (2AFC) protocol suggested in Kolkin et al. (2019); Park et al. (2020), where participants are shown

Table 1: We evaluate our method in temporal consistency by computing warp-error and conducting a user study, and in fidelity to the target text prompt using CLIP similarity. See Sec. 5 for more details.

| | Warp-err ↓ ($\times 10^{-3}$) | User preference of our method | CLIP score ↑ |
|---|---|---|---|
| LDM recon. | 2.0 | – | 0.23 |
| PnP-Diffusion | 11.3 | 94% | 0.33 |
| Text2Video-Zero | 12.5 | 78% | 0.33 |
| Tune-a-Video | 30.0 | 82% | 0.31 |
| Fate-Zero | 6.9 | 71% | 0.32 |
| Gen1 | – | 70% | 0.32 |
| Rerender-a-Video | 1.8 | 71% | 0.32 |
| Ours *w joint attention* | 5.9 | 90% | 0.33 |
| Ours *w/o rand keyframes* | 3.7 | – | 0.33 |
| Ours | 3.0 | – | 0.33 |

the input video, ours and a baseline result, and are asked to determine which video is more temporally consistent and better preserves the motion of the original video. The survey consists of 2000-3000 judgments per baseline obtained using Amazon mechanical turk. We note that warping-error could not be measured for Gen1 since their product platform does not output the same number of input frames. Table 1 compares our method to baselines. Our method achieves the highest CLIP

Figure 7: **Limitations.** Our method edits the video according to the feature correspondences of the original video, hence it cannot handle edits that requires structure deviations.

score, showing a good fit between the edited video and the input guidance prompt. Furthermore, our method has a low warping error, indicating temporally consistent results. We note that Re-render-a-Video optimizes for the warping error and uses optical flow to propagate the edit, and hence has the lowest warping error; However, this reliance on optical flow often creates artifacts and long-range inconsistencies which are not reflected in the warping error. Nonetheless, they are apparent in the user study, that shows users significantly favoured our method over all baselines in terms of temporal consistency. Additionally, we consider the reference baseline of passing the original video through the LDM auto-encoder without performing editing (*LDM recon.*). This baseline provides an upper bound on the temporal consistency achievable by LDM auto-encoder. As expected, the CLIP similarity of this baseline is poor as it does not involve any editing. However, this baseline does not achieve zero warp error either due to the imperfect reconstruction of the LDM auto-encoder, which hallucinates high-frequency information.

We further evaluate our correspondences and video representation by measuring the accuracy of video reconstruction using TokenFlow. Specifically, we reconstruct the video using the same pipeline of our editing method, only removing the keyframes *editing* part. Table 2 reports the PSNR and LPIPS distance of this reconstruction, compared to vanilla DDIM reconstruction. As seen, TokenFlow reconstruction slightly improves DDIM inversion, demonstrating robust frame representation. This improvement can be attributed to the keyframe randomization; It increases robustness to challenging frames since each frame is reconstructed from multiple other frames during the generation. Notably, our evaluation focuses on accurate correspondences within the feature space during generation, rather than RGB frame correspondences evaluation, which is not essential to our method.

## 5.3 ABLATION STUDY

First, we ablate the use of *TokenFlow*, Sec. 4.2, for enforcing temporal consistency. In this experiment, we replace *TokenFlow* with extended attention (Eq. 3) and compute it between each frames of the edited video and the keyframes (*w joint attention*). Second, we ablate the randomizing of the keyframe selection at each generation step (*w/o random keyframes*). In this experiment, we use the same keyframe indices (evenly spaced in time) across the generation. Table 1 (bottom) shows the quantitative results of our ablations, the resulting videos can be found in the SM. As seen, *TokenFlow* ensures higher degree of temporal consistency, indicating that solely re-

Table 2: We reconstruct the video using the TokenFlow pipeline, excluding keyframe editing. We evaluate the TokenFlow representation with PSNR and LPIPS metrics. Our reconstruction improves vanilla DDIM inversion, highlighting the robustness of TokenFlow representation.

|  | PSNR ↑ | LPIPS↓ |
|---|---|---|
| LDM recon. | 31.13 | 0.03 |
| DDIM inversion | 25.32 | 0.14 |
| **Ours** | **25.74** | **0.13** |

lying on the extension of self-attention to multiple frames is insufficient for achieving fine-grained temporal consistency. Additionally, fixing the keyframes creates an artificial partition of the video into short clips between the fixed keyframes, which reflects poorly on the consistency of the result.

## 6 DISCUSSION

We presented a new framework for text-driven video editing using an image diffusion model. We study the internal representation of a video in the diffusion feature space, and demonstrate that consistent video editing can be achieved via consistent diffusion feature representation during the generation. Our method outperforms existing baselines, demonstrating a significant improvement in temporal consistency. As for limitations, our method is tailored to preserve the motion of the original video, and as such, it cannot handle edits that require structural changes (Fig 7.) Moreover, our method is built upon a diffusion-based image editing technique to allow the structure preservation of the original frames. When the image-editing technique fails to preserve the structure, our method enforces correspondences that are meaningless in the edited frames, resulting in visual artifacts. Lastly, the LDM decoder introduces some high frequency flickering (Blattmann et al., 2023). A possible solution for this would be to combine our framework with an improved decoder (e.g., Blattmann et al. (2023), Zhu et al. (2023)). We note that this minor level of flickering can be easily eliminated with exiting post-process deflickering (see SM). Our work shed new light on the internal representation of natural videos in the space of diffusion models (e.g., temporal redundancies), and how they can be leveraged for enhancing video synthesis. We believe it can inspire future research in harnessing image models for video tasks, and for the design of text-to-video models.

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

Table 3: We report average runtime in seconds, of running ours and competing methods on a video of 40 frames.

| TAV | Text2video-zero | Rerender-a-video | fatezero | PnP | ours (preprocess) | ours (sampling) | ours (total) |
|---|---|---|---|---|---|---|---|
| 2684 | 198 | 285 | 349 | 208 | 50 | 187 | 237 |

We provide additional implementation details below. We refer the reader to the HTML file attached to our Supplementary Material for video results.

## A    IMPLEMENTATION DETAILS

**StableDiffusion.**    We use Stable Diffusion as our pre-trained text-to-image model; we use the *StableDiffusion-v-2-1* checkpoint provided via official HuggingFace webpage.

**DDIM inversion.**    In all of our experiments, we use DDIM deterministic sampling with 50 steps. For inverting the video, we follow Tumanyan et al. (2023) and use DDIM inversion with classifier-free guidance scale of 1 and 1000 forward steps; and extract the self-attention input tokens from this process similarly to Qi et al. (2023).

**Runtime.**    Since we don't compute the attention module on most video frames (i.e., we only compute the self-attention output on the keyframes) our method is efficient in run-time, and the sampling of the video *reduces the time of per-frame editing by 20%*. The inversion process with 1000 steps is the main bottleneck of our method in terms of run-time, and in many cases a significantly smaller amount of steps is suffieicent (e.g. 50). Table 3 reports runtime comparisons using 50 steps in all methods. Notably, our sampling time is indeed faster than that of per-frame editing (PnP).

**Hyper-parameters.**    In equation 5 we set $w_i$ to be:

$$w_i = \sigma(d_-/(d_+ + d_-))$$
$$\text{where } d_+ = ||i - i^+||, d_- = ||i - i^-|| \tag{6}$$

where $\sigma$ is a sigmoid function, $i^+$ and $i^-$ are the future and past neighboring keyframes of $i$, respectively.
For sampling the edited video we set the classifier-free guidance scale to 7.5. At each timestep, we sample random keyframes in frame intervals of 8. We note that using less keyframes (i.e., increasing the interval size) results in (i) runtime decreases (the joint keyframe editing step requires less memory, and is faster), and (ii) temporal consistency is improved (since the same tokens are shared across more frames). Nevertheless, too few keyframes will result in inaccurate correspondences which may result in artefacts.

**Baselines.**    For running the baseline of Tune-a-video (Wu et al., 2022) we used their official repository. For Gen-1 (Esser et al., 2023) we used their platform on Runaway website. This platform outputs a video that is not in the same length and frame-rate as the input video; therefore, we could not compute the warping error on their results. For text-to-video-zero (Khachatryan et al., 2023b) we used their official repository, with their depth conditioning configuration. For Fate-Zero (Qi et al., 2023) with used their official repository, and verified the run configurations with the authors.

