# OpenReview forum: "TokenFlow: Consistent Diffusion Features for Consistent Video Editing"
_ICLR.cc/2024/Conference — ICLR 2024 poster_

### Official Review · Reviewer_Mqtb · 2023-10-24

**Soundness:** 3 good
**Presentation:** 3 good
**Contribution:** 2 fair
**Rating:** 6
**Confidence:** 4

**Summary:**

In this paper, the authors propose a framework, TokenFlow, for video editing task.
TokenFlow runs in a correspondence-propagation manner, i.e., first seeks for the correspondences across different frames, jointly edits the keyframes, and then propagates features to ensure the temporal consistency.
Compared to prior arts, TokenFlow shows a better temporal consistency and competitive editing fidelity.

**Strengths:**

Approach:
- Global consistency. TokenFlow utilizes a joint editing and feature propagation via NN feature correspondences. Compared to attention-based methods, this way is more explicit and tends to keep a global consistency across different frames in a video.
- Compatible with other image-based editors. TokenFlow seems to be able to work with other diffusion-based image editors.

Experiments & validation:
- Proposed method is intuitive. First doing joint editing and then propagating the features makes sense.
- From qualitative results, TokenFlow improves the temporal consistency and preserves fair fidelity.
- Instead of using other pixelwise correspondences (e.g., dense flow or pixelwise trajectory), the authors propose to use nearest neighbor (NN) to find the correspondence. This seems new to me.

Writing & presentation:
- The paper is well-written and easy to follow.

**Weaknesses:**

Experiments:
- Compared to other baselines, like Text2LIVE and Gen1, TokenFlow still shows some "flickering" when there are high-frequency patterns. For example, in "Comparisons to Baselines" SM, in the first example "running dog", the ground has severe flickering compared to Gen 1. Also in the third example "cutting bread", there is more flickering in the bread and background compared to Gen 1. Same thing also happened in the comparison to Text2LIVE in "Additional Qualitative Comparisons". Why is TokenFlow not able to maintain the consistency for the high-frequency patterns?
- Based on the previous point, I think the authors could consider analyzing the reason behind the flickering and include it in **Limitations**.
- How do different image editors affect the results? Specifically,  the comparison w/ PnP + propagation. The authors mention that they have an additional comparison with PnP-Diffusion + propagation in the **last sentence, Section 5 Baselines**, but this part seems missing either in the main paper or in the SM.
- It would be great if the authors could also include runtime comparisons.

**Questions:**

- Features & RGB images: In Figure 3, authors show that TokenFlow improves the consistency in feature level. However, can the feature level consistency ensure the RGB output consistency?
- Correspondence ablation: Why using NN for the correspondences instead of using optical flow? Can pixel-level correspondences like dense optical flow be used for this token-based framework? Can we apply downsampled optical flow maps to the feature maps?
- Occlusion: Is the current framework be able to handle some extreme cases, like occlusion? For example, a dog running through some poles but sometimes the dog is occluded by pole.
- Video length: What is the maximal length TokenFlow can handle?
- What is the post-procssing deflickering that is used in the SM?

**Details Of Ethics Concerns:**

The proposed method might be used for human subject editing and could spread misinformation.

---

> ### Author Response · Authors · 2023-11-14
>
> Thank you for the supportive feedback.
> **High frequency patterns**
> As discussed in Sec 6 of our paper, the high frequency patterns in our results are attributed to the inherent nature of the latent diffusion model. Given that the latent space of the diffusion model is that of a VAE, which involves lossy compression, some high frequencies in the final RGB frames are hallucinated by its decoder. Such patterns are also evident when encoding-decoding a natural video without any further manipulations. We further discuss this limitation in Sec 6 of the paper.
> Different image editing techniques
> As discussed in the paper (Sec 5), our method can be combined with different structure-preserving image editing methods. All the results in the paper are using PnP as the image editing method; See our SM for results with ControlNet and SDEdit.
> **RGB propagation result**
> For the result of PnP editing + RGB propagation, please see our SM section “Additional Qualitative Comparisons” under the column “ebsynth” (the RGB propagation method); we are sorry if this title is misleading.
> **Consistency in features and in RGB**
> Our main premise in the paper is the correlation between feature and RGB consistency; our method shows that indeed, up to high frequencies that are hallucinated by the LDM decoder, feature consistency results in RGB consistency.
> **Video length**
> On an A100 it is possible to run ~300 frames. We believe some optimisations can be made to increase the amount of frames (for example, computing the extended attention on random subsets of the keyframes rather than on all of them).
> **Runtime**
> Please see our global response for runtime analysis.
> **Optical flow**
> Since our method performs propagation in the feature space, the swapped tokens should be ultimately interchangeable for the diffusion model. The most direct way to achieve this is through matching the features explicitly, rather than using subsampled pixel-level optical flow.
> **Post-process in SM**
> We used [this script](https://github.com/OndrejTexler/Few-Shot-Patch-Based-Training/blob/master/train.py).
> **Occlusions**
> Please see our global response.

---

### Official Review · Reviewer_6aWr · 2023-10-28

**Soundness:** 3 good
**Presentation:** 3 good
**Contribution:** 3 good
**Rating:** 8
**Confidence:** 4

**Summary:**

The paper aims to address the problem of consistent video editing. The proposed method is based on text-to-mage diffusion models, and the task is to convert a source video with a target text prompt into a new video that associates with the target text while preserving the motion of the source video. The emphasis is on producing consistent frames as naively applying an image-based text-to-image diffusion model would generate individually good-quality frames, but when they are put together, it would jointly result in an inconsistent video. The key idea proposed in this paper to solve the problem is called TokenFlow, which enforces the edited internal representation of the diffusion process to preserve the inter-frame correspondences of the original video. The approach is simple and the results shown in the paper and the supplementary material look quite good.

**Strengths:**

1.
The proposed method is simple and lightweight. It is built on an existing diffusion-based image editing method and does not need to fine-tune the model. The "TokenFLow Editing" algorithm is easy to implement (code available in the supplementary material). It directly utilizes Stable Diffusion, DDIM inversion, and PnP-Diffusion, and the TokenFlow procedure just requires computing the nearest-neighbor fields for token feature maps.
Further, as mentioned in the summary in Sec. 1 of the paper, state-of-the-art editing results are one of the main contributions of this work. Indeed, the edited videos presented in the supplementary results exhibit better consistency than other methods' outputs.

2.
A helpful finding from this work is that the internal features offer a shared and consistent representation across frames, and the corresponding features are interchangeable for the diffusion model. The spatial and semantic properties of diffusion features are also mentioned in the concurrent work "Emergent Correspondence from Image Diffusion" as DIFT, proposed by Tang et al.; however, they focus more on matching different images and show some results for edit propagation in image editing and for video label propagation on DAVIS and JHMDB instead of enforcing consistency in video editing. While it is not necessary to empirically compare TokenFlow with DIFT, it would be helpful to highlight the differences and the shared ideas.
Nevertheless, these findings of diffusion features provide a promising direction to revisit prior ideas like *Image Analogies* and *PatchMatch*.

**Weaknesses:**

1.
The results in the paper and the supplementary material mainly demonstrate the visual effect of video style transfer. For more general video editing tasks, one might expect to see some results of motion-based or composition-based video editing. Since the proposed method relies on the feature correspondences in the original video, it seems not trivial if one would like to modify the TokenFlow for motion-based editing.

2.
Regarding the quantitative evaluation:
- The *edit fidelity* measured by CLIP score does not provide useful/discriminative information. It might also need to include a user study on the visual quality and fidelity.
- The *temporal consistency* measured by optical flow and warping might over-penalize edits that change in shape and tend to favor edits that involve only color/texture changes.


3.
Minor typos:
-  "to operate on more **then** a single frame"
-  **keyframess'**
- The first words after Eqs. (4) \& (5) are in upper case: **Where**

**Questions:**

* What would happen if the target of editing is partially occluded for a few frames in the video?
* The supplementary material shows some results of per-frame editing using ControlNet. Is it possible for ControlNet to be used not only for editing but also for providing optical flow guidance? If so, how would it differ from TokenFLow?

---

> ### Author Response · Authors · 2023-11-14
>
> Thank you for the supportive feedback.
>
> **Diffusion Features - ongoing and future research**
> Thank you for the insightful discussion and ideas.
> The concurrent work DIFT is related to ours by studying the intermediate representation learned by a pre-trained text-to-image model. DIFT focuses on identifying features that can serve as localized visual descriptors for general visual tasks such as segmentation and image correspondences.  We focus our analysis on the properties of the features across video frames (e.g., interchangeability w.r.t. the diffusion model), and harness our findings, during the generation process, to expand the capabilities of a text-to-image model to consistent video editing. Leveraging our findings to revisit classical patch-based methods is an intriguing future research direction.
>
> **Structure and motion deviation**
> Please see our common comment to all reviewers for a discussion on structure and motion deviations.
>
> **Metrics**
> We wish to note that (a) measuring edit fidelity using CLIP score shows that our method is on par in terms of editing capabilities w.r.t. the baselines and other methods; we do not claim to improve visual quality w.r.t. per-frame editing (e.g., PnP); we will make an honest effort to further evaluate visual quality with a user study in the revised version. (b) Ours and the competing methods (besides TAV) are designed to preserve the structure of objects, hence we find this metric valid.  Note that in addition to this metric,  the paper includes a thorough user study to evaluate temporal consistency.
>
> **Occlusions**
> Please see our global response.
>
> **ControlNet and optical flow**
> We would appreciate further clarification of this question by the reviewer.

---

### Official Review · Reviewer_vSnb · 2023-10-31

**Soundness:** 3 good
**Presentation:** 3 good
**Contribution:** 2 fair
**Rating:** 6
**Confidence:** 4

**Summary:**

This paper proposes TokenFlow for text-driven video editing, aiming to generate a temporally consistent video that adheres to the text prompt while preserving the spatial structure/motion of the source video. Specifically, TokenFlow leverages a pre-trained text-to-image diffusion model to extract features/tokens of each video frame, compute latent patch correspondence between neighboring frames, and temporally propagate the key-frame tokens to other frames during the diffusion process. Qualitative and quantitative results show that the TokenFlow performs similarly to prior methods in terms of edit fidelity (CLIP similarity) while achieving higher temporal consistency (warping error and user study).

**Strengths:**

S1: Sensible model design
Although the ideas of 1) using text-to-image diffusion model for video generation and 2) using latent feature flow for temporal consistency are not new, the proposed framework combines these components sensibly. The simplicity of this method also makes it compatible with existing video editing methods and more efficient than most prior arts.

S2: Temporally consistent results
The visual results show a significant improvement from prior methods in terms of temporal consistency of both texture and structural details.

S3: Good writing
The paper is well-written and easy to follow. I find the illustrations and algorithm pseudo-code quite helpful to understand the framework.

**Weaknesses:**

W1: Novelty
The novelty of the proposed framework is slightly limited, considering that the key components (keyframe sampling, feature aggregation and propagation across frames) are introduced in prior works. Also, it is unclear which part of Section 4.1 is newly proposed in the paper and which is borrowed from other works. It would be great if the authors can elaborate on the main differences from prior methods and specify the novel components/modifications.

W2: Limited structural deviation
As shown in Figure 7, TokenFlow outputs strictly follow the structural layout of the source video, which might limit its generative capability/application. I’m wondering if there is a way to relax the temporal consistency constraint around object boundaries, so that one can find the desired tradeoff between temporal consistency and structural editing (maybe by tuning some hyper-parameters).

**Questions:**

Q1: The paper mentions that TokenFlow is more computationally efficient. What is the overall runtime to generate a new video and how is it compared to the methods listed in Table 1?

Q2: The ablation study on keyframe sampling only covers fixed and random sampling. It would be good to also ablate on sampling interval (tradeoff between computation overhead and temporal smoothness). I’m also curious if a dynamic keyframe sampling scheme would further improve the results, especially for occlusion cases.

---

> ### Author Response · Authors · 2023-11-14
>
> Thank you for the supportive feedback.
> **Differences w.r.t prior work**
> **To the best of our knowledge, we are the first to introduce a method that performs keyframe editing and propagation directly in the feature space of the model, during the generation process** (as pointed out by reviewers FeQk and Mqtb). Furthermore, our work provides new findings about diffusion features across video frames, supported by empirical analysis (as acknowledged by reviewers FeQk and 6aWr). On the technical side, our framework involves several key and novel components including: TokenFlow propagation – directly extracting and enforcing inter-frame correspondence in the feature space, and a dedicated generation process that alternates between keyframe editing and feature propagation.
>
> This is in contrast to prior work that either: (i) solely rely on extended attention (Sec. 4.1) , which is insufficient for temporal consistency, as we thoroughly demonstrated in Sec. 5, (e.g. [1], [2]) or (ii) edits keyframes relying on optical flow estimation between them, and propagates edits in RGB using an off-the-shelf propagation method (e.g., [3]); an approach which results in long range inconsistencies and artifacts (see our SM video comparisons), since optical flow estimation between distant frames (i.e. keyframes) is prone to errors.
>
> **The effect of interval size**
> We thank the reviewer for the interesting suggestion. The interval size is equivalent to changing the number of keyframes. We note that as less keyframes are used in our framework: (i) runtime decreases (editing less keyframe is faster and requires less memory), and (ii) temporal consistency is improved (since the same tokens are shared across more frames). Nevertheless, too few keyframes will result in inaccurate correspondences which may result in artefacts. Following this comment, we evaluated the effect of the interval size numerically on a subset of videos of 96 frames:
>
> | interval size | warp error (x 10^-3) | runtime |
> |---|---|---|
> | 4 | 2.3 | 1656 |
> | 8 | 2.2 | 656 |
> | 12 | 2.1 | 435 |
> | 16 | 2.0 | 359 |
>
>
>
> Empirically, we found in all our experiments that an interval size of 8 is a robust choice, balancing runtime and quality of the result.
>
> **Structure deviation**
> Please see our global response.
>
> **Occlusions**
> Please see our global response.
>
> [1]Chenyang Qi, Xiaodong Cun, Yong Zhang, Chenyang Lei, Xintao Wang, Ying Shan, and Qifeng Chen. Fatezero: Fusing attentions for zero-shot text-based video editing. arXiv:2303.09535, 2023.
>
> [2] Levon Khachatryan, Andranik Movsisyan, Vahram Tadevosyan, Roberto Henschel, Zhangyang Wang, Shant Navasardyan, and Humphrey Shi. Text2video-zero: Text-to-image diffusion models are zero-shot video generators. ArXiv, abs/2303.13439, 2023a.
>
> [3]  Shuai Yang, Yifan Zhou, Ziwei Liu, and Chen Change Loy. Rerender a video: Zero-shot text-guided video-to-video translation, 2023.

---

### Official Review · Reviewer_FeQk · 2023-11-01

**Soundness:** 3 good
**Presentation:** 4 excellent
**Contribution:** 3 good
**Rating:** 8
**Confidence:** 4

**Summary:**

The paper presents a method for text-based video editing, called TokenFlow. TokenFlow utilizes a pre-trained text-to-image diffusion model without the need for finetuning or video training data. Independently using text-based image editing techniques on frames will produce temporal artifacts. The paper proposed a method to improve the temporal consistency. More specifically, the method uses extended attention to edit several keyframes and then propagates the keyframe features to all the frames based on a Nearest Neighbour field. The Nearest Neighbour field is computed based on features of DDIM inversion.

**Strengths:**

- The video editing results are impressive, the temporal consistency is pretty good.
- The analysis and visualization of UNet features on video tasks are helpful for future research on video generation.
- The idea of TokenFlow is novel. Based on the ablation study and qualitative results in the supplemental material, TokenFlow is also very critical to good temporal consistency.
- The paper reads well and is easy to follow.

**Weaknesses:**

Although it's not necessary, it will be helpful to compare TokenFlow with Pix2Video.

**Questions:**

Are self-attention features the only features that are replaced by features of neighboring frames? Have you tried to replace some other features such as ResBlock features or attention masks?

---

> ### Author Response · Authors · 2023-11-14
>
> Thank you for the supportive feedback.
> **Feature selection**
> It is indeed only the self attention tokens that are being replaced. We selected these features as they have been shown to capture fine-grained spatial information, and have been used for controlled image generation [1]. This is in contrast to the cross attention maps that capture only rough layout and hence would not be a good fit for our goal of fine-grained temporal correspondence [2].
> Note that the residual block is followed by a self attention block in every layer, thus the residual features are also overridden (up to skip connections) by our token replacement.
>
>
> [1] Narek Tumanyan, Michal Geyer, Shai Bagon, and Tali Dekel. Plug-and-play diffusion features for text-driven image-to-image translation. Proceedings of the IEEE/CVF Conference on Computer Vision and Pattern Recognition (CVPR), June 2023.
> [2] Hertz, Amir, Ron Mokady, Jay Tenenbaum, Kfir Aberman, Yael Pritch, and Daniel Cohen-Or. Prompt-to-prompt image editing with cross attention control. ICLR (2022).$

---

### Author Response · Authors · 2023-11-14

We thank you for these thorough reviews.
There were a couple of mutual points to all reviewers we wish to address here:
**Structure deviations and Motion editing**
Our method is designed to preserve the motion and object structure of the original video. Supporting video edits that involve significant structure deviations or editing of movements of objects, would require a motion prior to allow such edits to be applied in a consistent manner across frames.  For example, changing a wolf to a rabbit would require synthesizing new motion of the ears, which is not present in the original video. It is unclear how an image diffusion model can provide such motion information. With the rapid progress in text-to-video diffusion models, we anticipate that generative motion priors suitable for these editing tasks will become available, and could be used to extend our work.

**Runtime evaluation**
The table below reports average runtime comparisons on a video of 40 frames, using 50 diffusion steps in all methods. Notably, our sampling time is indeed faster than that of per-frame editing (PnP).
| TAV | Text2video-zero | Rerender-a-video | fatezero  | PnP | ours (preprocess) | ours (sampling) | ours (total)  |
|---|---|---|---|---|---|---|---|
| 2684.702563 | 198.5167881 | 285.848508 | 349.0715174 | 208.4318189 | 50.6622076 | 187.204452 | 237.8666596  |
| 578.8067357 | 4.807203365 | 26.62864091 | 26.86996169 | 2.176882622 | 11.11061152 | 3.114096461 | 9.716412365 |

**Occlusions**
Thanks to the random sampling of keyframes in our method, it is quite robust to occlusions, since it is highly unlikely that an object would be occluded in the samples keyframes in all timesteps. We empirically didn’t observe that occlusions are an issue.

---

### Meta-Review · Area_Chair_bqXb · 2023-12-09

**Metareview:**

This paper tackles text-guided video editing problem using a pre-trained text-to-image diffusion model with a focus on temporal consistency.

Reviewers recognized the simplicity and intuitiveness of the proposed method, and the effectiveness in temporal consistency and fidelity preservation. Simultaneously, weaknesses in terms of flickering artifact for high-frequency patterns and the missing comparisons from interchangeable image diffusion models. The authors agreed to provide extra discussion on the limitations and the experimental results.

I would recommend acceptance of this paper.

**Justification For Why Not Higher Score:**

There were several imitations pointed out by the reviewers and the authors admitted it. This paper has both strengths and weaknesses.

**Justification For Why Not Lower Score:**

While the limitations exist, the strengths recognized by the reviewers outweighs the limitation.

---

### Decision · Program_Chairs · 2024-01-16

Accept (poster)